

# Insect herbivores increase mortality and reduce tree seedling growth of some species in temperate forest canopy gaps

Nathan P. Lemoine[1], Deron E. Burkepile[2] and John D. Parker[3]

[1] Department of Biology, Colorado State University, Fort Collins, CO, United States
[2] Department of Ecology, Evolution, and Marine Biology, University of California, Santa Barbara, CA, United States
[3] Smithsonian Environmental Research Center, Edgewater, MD, United States

## ABSTRACT

Insect herbivores help maintain forest diversity through selective predation on seedlings of vulnerable tree species. Although the role of natural enemies has been well-studied in tropical systems, relatively few studies have experimentally manipulated insect abundance in temperate forests and tracked impacts over multiple years. We conducted a three-year experiment (2012–2014) deterring insect herbivores from seedlings in new treefall gaps in deciduous hardwood forests in Maryland. During this study, we tracked recruitment of all tree seedlings, as well as survivorship and growth of 889 individual seedlings from five tree species: *Acer rubrum*, *Fagus grandifolia*, *Fraxinus* spp., *Liriodendron tulipifera*, and *Liquidambar styraciflua*. Insect herbivores had little effect on recruitment of any tree species, although there was a weak indication that recruitment of *A. rubrum* was higher in the presence of herbivores. Insect herbivores reduced survivorship of *L. tulipifera*, but had no significant effects on *A. rubrum*, *Fraxinus* spp., *F. grandifolia*, or *L. styraciflua*. Additionally, insects reduced growth rates of early pioneer species *A. rubrum*, *L. tulipifera*, and *L. styraciflua*, but had little effect on more shade-tolerant species *F. grandifolia* and *Fraxinus* spp. Overall, by negatively impacting growth and survivorship of early pioneer species, forest insects may play an important but relatively cryptic role in forest gap dynamics, with potentially interesting impacts on the overall maintenance of diversity.

Corresponding author
Nathan P. Lemoine,
lemoine.nathan@gmail.com

## INTRODUCTION

Insect herbivores can directly and indirectly influence plant community composition by altering the recruitment, mortality, or individual growth rates of plant species (*Maron & Crone, 2006*; *Kim, Underwood & Inouye, 2013*). In old fields, for example, insect herbivores indirectly increase the cover of subdominant plant species by reducing the growth of competitively superior species (*Brown & Gange, 1992*; *Davidson, 1993*; *La Pierre, Joern & Smith, 2015*). In tropical forests, insect herbivores increase the diversity of tree seedling recruits (*Dyer et al., 2010*; *Swamy & Terborgh, 2010*; *Terborgh, 2012*; *Bagchi et al., 2014*). In temperate forests, however, the most well-documented and obvious impacts of insects come from pest species outbreaks, like mountain pine beetles, which cause widespread

mortality of dominant tree species (*Romme, Knight & Yavitt, 1986*). However, we know relatively little about the role of insects in temperate forests during non-outbreak scenarios (*Maron & Crone, 2006*), because there have been few experimental manipulations of non-outbreaking insect herbivores in temperate forests.

The first pathway by which herbivorous insects might influence forest dynamics is altered germination and emergence (hereafter termed recruitment). Unfortunately, few data exist regarding the effects of herbivores on seedling emergence. Recruitment is suppressed by herbivores for many tree species via predation on newly germinated seedlings (*Meiners, Handel & Pickett, 2000*; *Dulamsuren, Hauck & Mühlenberg, 2008*). Reduced germination, caused by insect herbivores, can alter forest successional trajectories over long time scales by modifying initial community composition (*Bagchi et al., 2014*). Indeed, fast-growing pioneer species are often the most susceptible to herbivory (*Coley, Bryant & Chapin, 1985*). Insect herbivores can also increase spatial heterogeneity in species composition by restricting the locations in which particular species can establish (*Fine, Mesones & Coley, 2004*; *Bagchi et al., 2014*). Here, we hypothesized that insect herbivores will reduce recruitment of palatable, fast-growing tree species, potentially increasing community diverse over longer time scales.

Insect herbivores can also influence forest dynamics via mortality of emerged and established seedlings. *Norghauer & Newbery (2013)* and *Norghauer & Newbery (2014)*, for example, demonstrated that herbivorous insects significantly alter forest community composition through selective predation on particular seedling species. Such selective predation can allow for coexistence of competitively inferior seedlings by reducing growth rates of competitively superior species (*Fine, Mesones & Coley, 2004*; *Fine et al., 2006*). This mechanism is thought to be at least partly responsible for the maintenance of forest diversity (*Swamy & Terborgh, 2010*; *Terborgh, 2012*; *Bagchi et al., 2014*). Insect herbivores can also cause immediate plant death, tripling normal seedling mortality rates (*Prittinen et al., 2003*), whereas in other cases insect-caused seedling mortality may occur up to several years following initial defoliation (*Eichhorn et al., 2010*). Similarly, chronic foliar damage can exacerbate competition-induced mortality among seedlings (*Meiners & Handel, 2000*), highlighting the need for relatively long-term studies of insect impacts on tree seedling dynamics. We hypothesized that early pioneer species, being the most palatable, would suffer the highest mortality rates of common species in a temperature forest.

Finally, herbivores reduce growth of numerous tree species. In particular, fast-growing, palatable species suffer the greatest reduction in growth following herbivory (*Fine, Mesones & Coley, 2004*); whereas less-palatable, slow growing species are often more tolerant of herbivory (*Siemann & Rogers, 2003*). Indeed, herbivores can equalize growth rates among species that vary in competitive ability (*Norghauer & Newbery, 2013*) and subdominant species often only persist in communities with insect herbivores (*Brown & Gange, 1992*; *Davidson, 1993*; *La Pierre, Joern & Smith, 2015*). Herbivory can, therefore, restrict species to particular habitats or soil types, ultimately increasing beta diversity in tropical forests (*Fine, Mesones & Coley, 2004*; *Fine et al., 2006*). We tested the hypothesis that insect herbivores would reduce growth of a subset of early successional, pioneer species whereas slower growing, late successional species would be more tolerant of insect herbivory.

The effects of insects on tree seedlings might be particularly strong within treefall gaps. Treefall gaps maintain forest diversity by increasing light availability and nitrogen mineralization rates (*Denslow, 1987*; *Mladenoff, 1987*; *Hubbell et al., 1999*), leading to abundant germination and recruitment of a diverse assemblage of tree species (*Denslow, 1987*). In addition, insect herbivores are attracted to the warmth and light in treefall gaps, increasing local abundances and thus their potential impacts on seedling growth and mortality (*Richards & Windsor, 2007*; *Norghauer & Newbery, 2013*). Moreover, early successional plant species are generally thought to possess high growth rates at the expense of anti-herbivore defenses, which might make them particularly susceptible to insect herbivory (*Coley, Bryant & Chapin, 1985*; *Shure & Wilson, 1993*). Thus, recruitment in forest gaps may be partially mediated by insect herbivore consumption of palatable new seedlings, but little information exists regarding the role of insect herbivores in forest regeneration in treefall gaps.

Here, we conducted a three-year experiment to assess the influence of insect herbivores on community dynamics within treefall gaps in a temperate forest in the eastern United States. We used a paired design, where each treefall gap consisted of two plots: a control plot and a plot sprayed with pesticide to remove insect herbivores. We then tracked seedling recruitment, mortality, and growth in each plot for three years.

## MATERIALS AND METHODS

We conducted this experiment in an intensively studied forest at the Smithsonian Environmental Research Center (SERC; Edgewater, Maryland USA. 38°53′N, 76°33′W). SERC comprises a 2,650 acre protected research facility encompassing a variety of early-, mid-, and late-successional forests. Most forests at SERC, including those in this study, are mid-successional forests 75–120 years old that have been federally protected since the 1960s (*Parker et al., 2010*). This forest was typical of those in the mid-Atlantic United States, dominated by *Carya* spp., *Fagus grandifolia*, *Fraxinus* spp., *Liquidambar styraciflua*, *Liriodendron tulipifera*, and *Quercus* spp (*Brush, Lenk & Smith, 1980*; *McMahon, Parker & Miller, 2012*). Insect communities in hardwood forests of North America are dominated by only a few groups of insects: diptera, hymenoptera, homoptera, coleoptera, and lepidoptera (*Rohr, Mahan & Kim, 2009*). Of these groups, roughly 20% individuals are phytophagous (*Ingwell et al., 2012*).

In May 2012, we surveyed the forest for new treefall gaps. We only recorded gaps that appeared during the previous winter, restricting our experiment to new forest regeneration. We found six new treefall gaps large enough to include two $1.5 \times 1.5$ m experimental plots. This plot size is within the range of sizes common in studies of insect herbivory (*Brown & Gange, 1992*; *La Pierre, Joern & Smith, 2015*). In May 2013, we added one additional gap that opened during the winter of 2012–2013, resulting in seven total gaps. All gaps were interspersed within a forest area of approximately 16 ha. In each gap we established two 2.25 m$^2$ plots, each entirely surrounded by 1 m high chicken wire to exclude deer, as we were solely interested in the impacts of insects in this experiment. Previous experiments at SERC have used identical plot designs to successfully eliminate deer browsing, where

deer densities are low (3–8 individuals km$^{-2}$) compared to many parts of the United States (*Cook-Patton, LaForgia & Parker, 2014*), and no evidence of deer browsing was observed in any plot over the course of this experiment. Plots within each gap were located at least 5 m apart, minimizing the possibility of pesticide spray drifting between plots. Plots were not cleared or weeded in order to leave all natural vegetation intact.

Within each gap, one plot was randomly designated as the 'Control' treatment and the other was designated as a 'Pesticide' treatment. 'Pesticide' plots were sprayed every 2–3 weeks with 0.5 L of ASANA XL (DuPont) pesticide, diluted to a concentration of 40 μL/L. ASANA XL is an esfenvalerate insecticide that contains very little nitrogen, such that fertilization effects on plant growth should be minimal. Furthermore, ASANA XL is highly resistant to both rainfall and UV degradation and remains as a highly effective residue on leaf surfaces for several weeks after the initial application (*DuPont, 2006*). It is therefore commonly used to examine the effects of insect herbivory on plant communities (*Siemann & Rogers, 2003*; *Heath et al., 2014*).

Control plots were sprayed with an equivalent amount of water to control for possible watering effects. Pesticide applications began each year in May and continued through October, encompassing the entire growing season. Pesticide was applied for three years, beginning in 2012 and ending in 2014. We sprayed the entirety of each plot, including ground cover, in order to judge how insects affect recruitment. To judge the efficacy of the pesticide treatment, we surveyed each seedling in each plot for insect damage at the end of the 2012 growing season. We examined the top four leaves of each seedling, categorizing damage as 0%, 25%, 50%, 75%, or 100% damage and then averaged the values over all seedlings in each plot (*Palmer & Brody, 2013*; *Johnson, Bertran & Turcotte, 2016*).

In May of each year, each seedling was identified, measured for initial height, and tagged with a unique identification number. In October of each year, plots were censused again to record seedling final height, mortality, and recruitment over the growing season. Seedlings were only recorded as dead if an empty tag was found within the plot. New seedlings without tags were recorded as new recruits in both May and October, although only recruitment over the growing season (i.e., recruits appearing in October) was examined here, as recruits over the winter were not exposed to pesticide treatment. Seedlings were considered dead if they had no measurements for at least two consecutive census periods and did not reappear in the plot at a later date. In total, we tagged and measured 1,173 seedlings over the course of the experiment.

## Data analysis

We examined pesticide effects only on seedlings ≤15 cm in height because these were the newest recruits and thus the most likely to experience mortality from herbivory. The term seedling has a broad and inconsistent usage, referring to plants anywhere between zero and four years old (*Hanley et al., 2004*), but we use seedling to refer any individual ≤15 cm in height. Of the original dataset, 924 individuals (78%) met this criterion. We further restricted our analyses to the five most common species that occurred in enough plots to provide reasonable replication at the species level: *Acer rubrum*, *Fagus grandifolia*, *Fraxinus* spp., *Liriodendron tulipifera*, and *Liquidambar styraciflua* (Table 1). These five species
**Table 1** Numbers of seedlings (individuals < 15 cm tall) observed for each species during each year for each of the five focal species at the Smithsonian Environmental Research Center.

| Species | Year 1 2012 | Year 2 2013 | Year 3 2014 |
|---|---|---|---|
| *Acer rubrum* | 29 | 23 | 22 |
| *Fagus grandifolia* | 22 | 24 | 30 |
| *Fraxinus* spp | 42 | 41 | 28 |
| *Liriodendron tulipifera* | 332 | 278 | 191 |
| *Liquidambar styraciflua* | 55 | 51 | 14 |

comprised 82% of observed seedlings (76% of all tagged individuals) and are dominant species in the eastern United States (*Parker, O'Neill & Higman, 1989*).

We conducted all analyses using Bayesian methods, which allowed us to incorporate prior information that prevented us from overestimating effect sizes. Small sample sizes, as used here ($n = 7$ gaps), often lead to overestimates of the true effect sizes when using traditional analytical methods. Such overestimates have been a persistent problem in biological research (*Button et al., 2013*). Bayesian methods allowed us to place weakly informative priors $N(0,1)$ on all parameters, which shrink parameter estimates towards 0 and help prevent overestimating effect sizes during the analysis of small sample sizes (*Kruschke, 2010*; *Button et al., 2013*).

We analyzed total recruitment using Bayesian hierarchical Poisson regression, with species as random effects (Appendix 1). Because recruitment in any given year was absent for many species, we calculated the total number of new recruits for each of the five species in each plot to yield total number of recruits observed for each species over the entire experimental duration.

We used a hierarchical Bayesian model to examine the influence of pesticide treatment and gap age on seedling mortality. We analyzed mortality rate using logistic regression that included pesticide treatment, gap age, and the interactions of these two categorical variables as predictors (Appendix 1). We did not use a paired model because not all species were present in all gaps during each year, yielding an unbalanced design. This limitation means that we cannot ascribe some variance in the response to spatial variation among treefall gaps, thereby making it more difficult to detect effects of insect herbivores. Plant species were random effects, with species-level coefficients varying around overall coefficients. Overall coefficients describe the aggregate influence of pesticide and gap age on overall seedling mortality. The correlation matrix between species-level parameters was given an uninformative Wishart prior. Variances of mortality rate were allowed to differ among species, relaxing the assumption of homogenous variances among species.

We analyzed relative growth rate [$100*(\text{Height}_{end} - \text{Height}_{start})/\text{Height}_{start}$] using a similar hierarchical Bayesian model as described above (Appendix 1). To account for the fact that individual seedlings were not independent within a plot, growth of individual seedlings within a plot varied around the mean growth of their respective species in that particular plot and year. The plot-level means were a function of pesticide treatment, gap age, and their interactions; species-level coefficients varied around the overall coefficients as described above (Appendix 1).
Posterior distributions for all parameters were constructed using 5,000 'burn-in' iterations that were discarded followed by 5,000 sampling iterations for each chain, saving every 10th posterior draw (four chains, 2,000 estimates per parameter). Response variables, except total recruitment, were standardized prior to analyses by subtracting the mean from each observation and dividing by the standard deviation in order to allow the use of weakly informative priors as described above. We examined model assumptions and fit using standardized residual plots. We report all statistics as probability of an effect (Pr), where higher probabilities denote more certainty regarding the importance of an effect. For example, $Pr = 0.95$ indicates that 95% of a coefficient's posterior distribution lay above or below zero. There is therefore a 95% chance that the coefficient is either negative or positive and therefore important. In contrast, $Pr = 0.50$ indicates that a coefficient is equally likely to be either positive or negative and therefore unimportant. To facilitate interpretation of probabilities, we provide effect sizes and 95% credible intervals for all effects. Credible intervals were calculated as the interval between the 2.5% and 97.5% quantiles for each posterior distribution (all posterior distributions were normal and symmetric). All analyses were conducted in Python v2.7. Bayesian models were built using STAN accessed via Pystan (*Stan Development Team, 2015*).

## RESULTS

At the beginning of the experiment in May 2012, plots contained an average of 20 seedlings each (range: 9–163 seedlings per plot). The five common species analyzed here accounted for ∼80% of all initially tagged seedlings, with *L. tulipifera* being the most common (36.3%) followed by *Fraxinus* spp. (15.8%), *F. grandifolia* (15.1%), *L. styraciflua* (8.5%), and *A. rubrum* (8.5%) (Table 1). Pesticide application reduced foliar damage by approximately half. Percent of leaf area removed in 'Control' plots was 7.38 ± 0.83%, whereas leaf area removal in 'Pesticide' plots was 4.27 ± 0.76% (Pr(Control > Pesticide) = 0.99). *Fagus grandifolia* suffered the highest insect damage (16.5 ± 5.7%), followed by *A. rubrum* (9.7 ± 5.9%), *L. styraciflua* (5.6 ± 2.7%), *Fraxinus* spp. (3.7 ± 3.7%), and *L. tulipifera* (0.64 ± 0.34%).

### Recruitment

In general, recruitment over the duration of the experiment was relatively low. Most species averaged fewer than two recruits during the growing season per plot over the duration of the experiment. *Liriodendron tulipifera* had the most recruits of any species, with plots averaging > 10 new recruits over the entire three years (Fig. 1). Surprisingly, long-term recruitment of two species, *A. rubrum* and *Fraxinus* spp., was higher in control plots compared to plots sprayed with pesticide (Pr(Control > Pesticide) ≥0.95 for both species) (Fig. 1, Table 2). Pesticide application did not affect the recruitment of the other three species (Pr(Control < Pesticide) <0.84 for all species) (Fig. 1 and Table 2).

### Mortality

Pesticide had negligible effects on seedling mortality averaged across all species (Pr(Control >Pesticide) = 0.66). However, species varied in their responses to pesticide. Pesticide did not

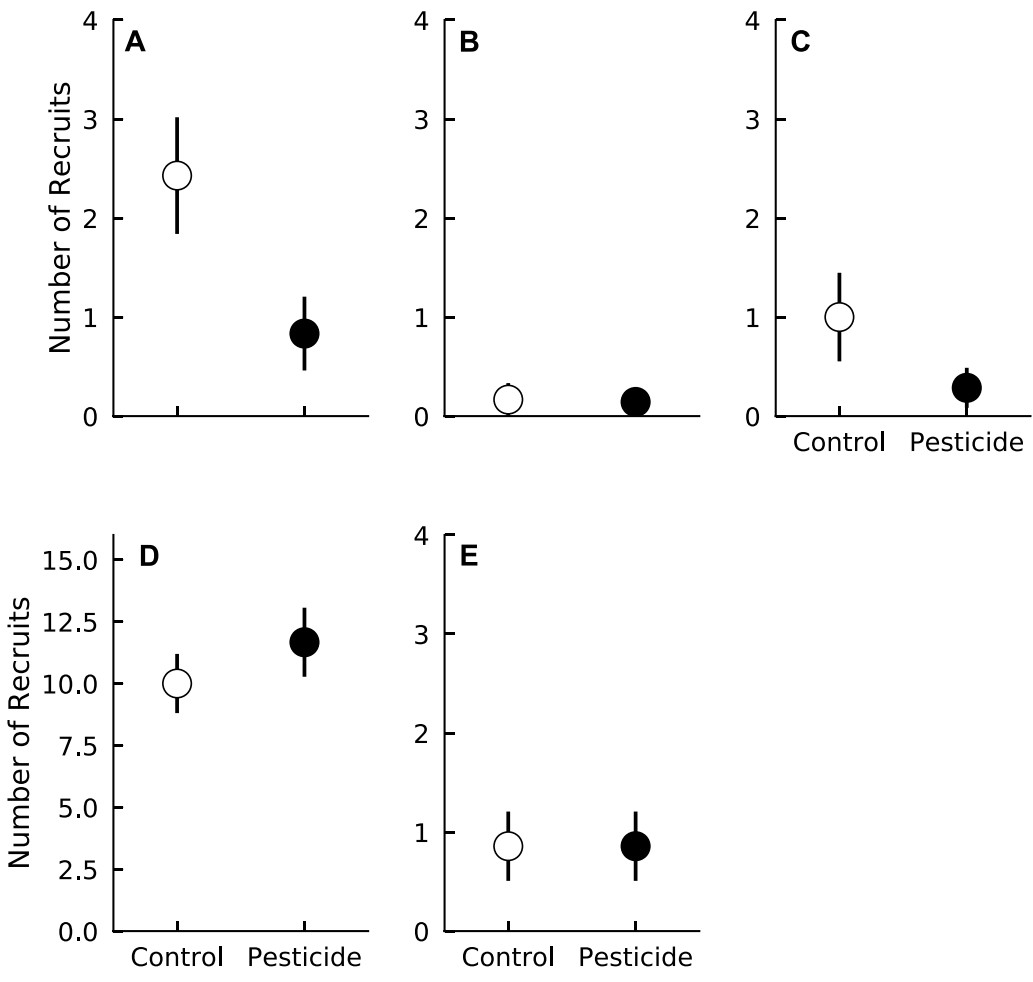

**Figure 1** **Number of recruits of five tree species in plots exposed to and protected from insect herbivores at the Smithsonian Environmental Research Center (Edgewater, MD) from 2012 to 2014.** (A) *Acer rubrum*, (B) *Fagus grandifolia*, (C) *Fraxinus spp.*, (D) *Liriodendron tulipifera*, (E) *Liquidambar styraciflua*. Points show means ± 1 SE. Data were pooled by each plot ($n = 7$ plots per treatment) prior to calculating treatment means and standard errors.

**Table 2** **Probability that each recruitment coefficient is greater or less than zero for each of the five species analyzed here.** Probabilities denote max(Pr(coefficient) > 0, Pr(coefficient) < 0), such that sign of the coefficient was ignored and the probabilities simply represent the probability of the coefficient being important in the model. Bold denotes Pr(coefficient) $\geq$ 0.90.

| Recruitment coefficient | *Acer rubrum* | *Fagus grandifolia* | *Fraxinus spp.* | *Liriodendron tulipifera* | *Liquidambar styraciflua* |
|---|---|---|---|---|---|
| Pesticide | **0.99** | 0.74 | **0.95** | 0.82 | 0.58 |

alter mortality rates of *A. rubrum*, *F. grandifolia*, *Fraxinus* spp., or *L. styraciflua* (Pr(Control >Pesticide) $\leq$0.79 for all four species, Fig. 2 and Table 3). In contrast, pesticide application marginally reduced mortality for *L. tulipifera* (Pr(Control >Pesticide) = 0.93) (Fig. 2, Table 3). Mortality of *L. tulipifera* was 33.1% higher in plots without pesticide compared to control plots (CI$_{95}$ = 9.9% lower–76.3% higher). Weak interaction coefficients suggest that

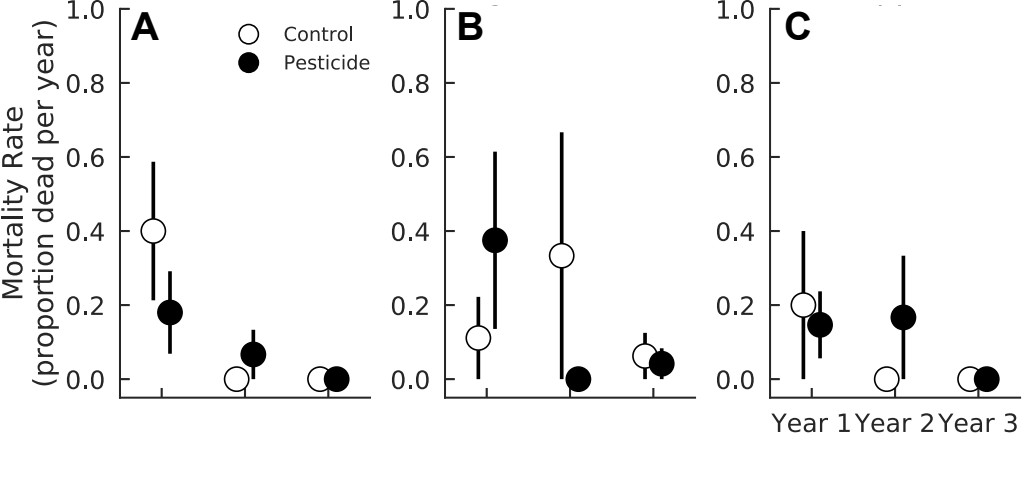

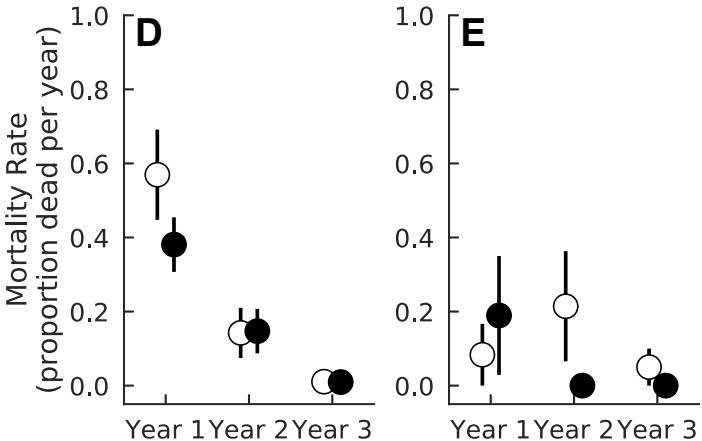

**Figure 2** **Mortality rate of five tree species in plots exposed to and protected from insect herbivores at the Smithsonian Environmental Research Center (Edgewater, MD) from 2012 to 2014.** (A) *Acer rubrum*, (B) *Fagus grandifolia*, (C) *Fraxinus spp.*, (D) *Liriodendron tulipifera*, (E) *Liquidambar styraciflua*. Points show means $\pm$ 1 SE. We calculated the proportion of mortality of each species in each plot ($n = 7$ plots per treatment) prior to calculating treatment means and standard errors.

pesticide effects were consistent across all years for all species (Table 3). Insects therefore marginally decreased survival of one common, temperate tree species, *L. tulipifera*.

Overall, total seedling mortality declined between the one and three year old gaps (Pr(Year 3 < Year 1) = 1.0), and this effect was consistent across all species except *L. styraciflua* (Table 3). Mortality of *F. grandifolia* and *L. styraciflua* marginally declined as gaps aged (Table 3). *Fraxinus* spp. mortality significantly decreased between one and three year old gaps (Pr(Year 3 > Year 1) = 0.98, Table 3). Mortality of *A. rubrum* and *L. tulipifera* declined rapidly between the first and second year (Table 3 and Fig. 2) and between the second and third year for *L. tulipifera* (Pr(Year 3 < Year 1) = 1.00). Mortality did not decline further between two and three year old gaps for *A. rubrum* (Pr(Year 3 < Year 2) = 0.88) (Fig. 2).

**Table 3  Probability that each mortality coefficient is greater or less than zero for each of the five species analyzed here.** Probabilities denote max(Pr(coefficient) > 0, Pr(coefficient) < 0), such that sign of the coefficient was ignored and the probabilities simply represent the probability of the coefficient being important in the model. Bold denotes Pr(coefficient) ≥ 0.90.

| Mortality coefficient | *Acer rubrum* | *Fagus grandifolia* | *Fraxinus* spp. | *Liriodendron tulipifera* | *Liquidambar styraciflua* |
|---|---|---|---|---|---|
| Pesticide | 0.79 | 0.68 | 0.65 | **0.93** | 0.56 |
| Year 2 | **0.99** | **0.93** | **0.97** | **1.00** | **0.91** |
| Year 3 | **0.99** | **0.93** | **0.98** | **1.00** | 0.87 |
| Pesticide × Year 2 | 0.63 | 0.84 | 0.71 | 0.78 | 0.85 |
| Pesticide × Year 3 | 0.66 | 0.63 | 0.72 | 0.59 | 0.70 |

**Table 4  Probability that each growth coefficient is greater or less than zero for each of the five species analyzed here.** Probabilities denote max(Pr(coefficient) > 0, Pr(coefficient) < 0), such that sign of the coefficient was ignored and the probabilities simply represent the probability of the coefficient being important in the model. Bold denotes Pr(coefficient) ≥ 0.90.

| Growth coefficient | *Acer rubrum* | *Fagus grandifolia* | *Fraxinus* spp. | *Liriodendron tulipifera* | *Liquidambar styraciflua* |
|---|---|---|---|---|---|
| Pesticide | **0.99** | 0.77 | 0.84 | 0.84 | **0.98** |
| Year 2 | 0.87 | 0.76 | 0.71 | **0.99** | **0.93** |
| Year 3 | 0.79 | 0.51 | 0.67 | 0.54 | 0.58 |
| Pesticide × Year 2 | **0.94** | **0.91** | **0.94** | **0.92** | **0.97** |
| Pesticide × Year 3 | **0.90** | 0.75 | 0.67 | 0.84 | 0.72 |

### Relative growth rates

Exposure to insects slightly decreased overall relative growth rates of seedlings (Pr(Contro < Pesticide) = 0.91), and this effect also varied among species (Table 4). *Acer rubrum* grew 23.7% faster (CI$_{95}$ = 4.1%–45.6%) in one-year-old gaps sprayed with pesticides (Pr(Control < Pesticide) = 0.99, Fig. 3). However, pesticide did not affect *A. rubrum* growth rates in either two or three year old gaps (Pr(Control < Pesticide) <0.69 for both years, Fig. 3, Table 4). Herbivory had little effect on relative growth rates of *F. grandifolia* (Pr(Control < Pesticide) <0.80 for all years) or *Fraxinus* spp. (Pr(Control < Pesticide) <0.85 for all years) (Fig. 3, Table 4). *Liriodendron tulipifera* growth increased markedly in two year old gaps (Pr(Year 2 > Year 1) = 1.00) but was unaffected by pesticide (Pr(Control < Pesticide) <0.85 for all years, Fig. 3 and Table 4). Insects suppressed growth of *L. styraciflua* by 19.3% (CI$_{95}$ = 1.3%–38.2%) in one and three year old gaps (Pr(Control < Pesticide) = 0.998). In two year old gaps, insects had no detectable effect on *L. styraciflua* growth rates (Pr(Control < Pesticide) = 0.33 in year two).

### DISCUSSION

Insects can kill >50% of canopy trees during outbreak years (*Romme, Knight & Yavitt, 1986*). Even during non-outbreak years, insect herbivores play important roles in forest ecosystem function. For example, over one quarter of annual nitrogen deposition in tropical forests derives from insect frass (*Metcalf et al., 2014*). Insect herbivores can also govern community composition on decadal time scales by changing the abundance and

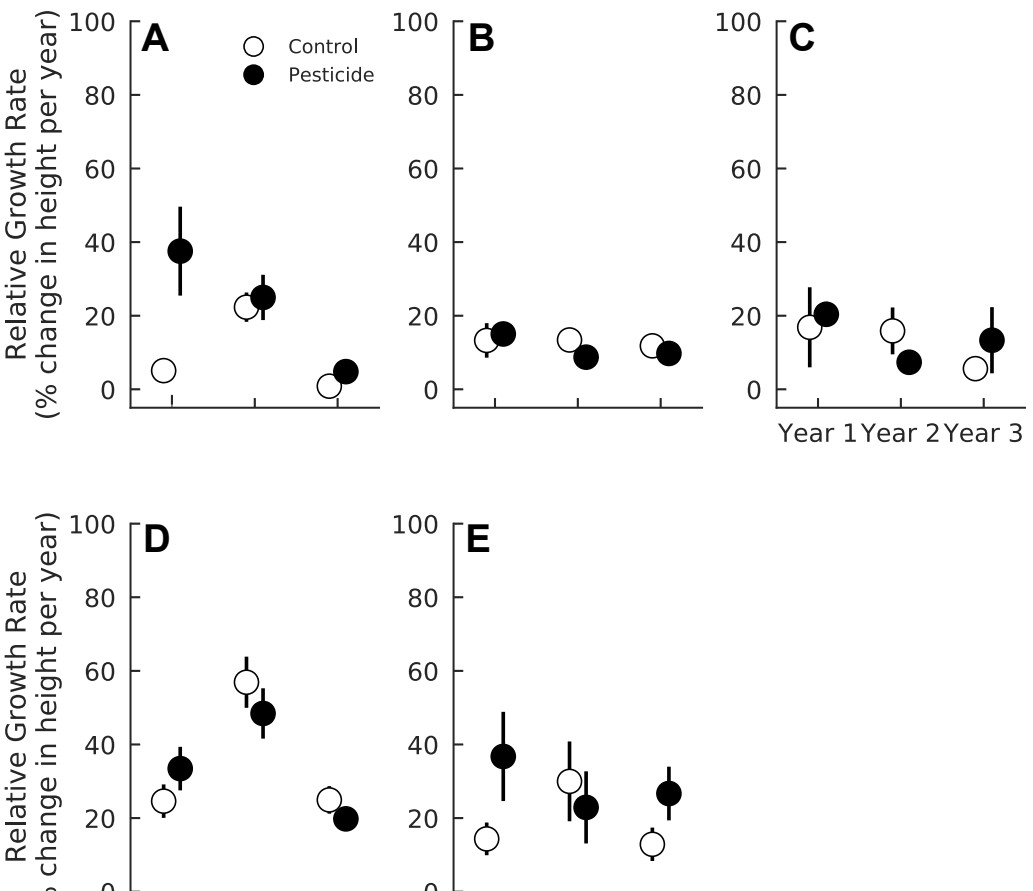

**Figure 3** Relative growth rates of five tree species in plots exposed to and protected from insect herbivores at the Smithsonian Environmental Research Center (Edgewater, MD) from 2012 to 2014. (A) *Acer rubrum*, (B) *Fagus grandifolia*, (C) *Fraxinus spp.*, (D) *Liriodendron tulipifera*, (E) *Liquidambar styraciflua*. We calculated mean growth rate of each species within each plot ($n = 7$ plots per treatment) prior to calculating treatment means and standard errors.

diversity of tree seedling recruits in tropical forests (*Dyer et al., 2010*; *Swamy & Terborgh, 2010*; *Terborgh, 2012*; *Bagchi et al., 2014*). In contrast, we know comparatively little about the role of insect herbivores in temperature forest regeneration in non-outbreak years. Our study demonstrates that insects play a relatively minor role in structuring temperate forest communities. Most seedling species were resistant or tolerant of insect herbivores, as insects decreased growth of only three early pioneer tree species (*A. rubrum, L. styraciflua, L. tulipifera*) and decreased survival of only *L. tulipifera* during the initial phase of forest regeneration in temperate treefall gaps.

Contrary to our initial hypotheses, recruitment of most species (*F. grandifolia, L. tulipifera*, and *L. styraciflua*) was unaffected by insects. Surprisingly, pesticide application appeared to reduce recruitment of both *A. rubrum* and *Fraxinus* spp, albeit weakly. These patterns are similar to those reported by *Meiners, Handel & Pickett (2000)*, where the presence of insect herbivores occasionally resulted in higher germination rates of *A. rubrum*.

In our study, increased cumulative recruitment of *A. rubrum* exposed to insects was largely driven by recruitment during the last year, in which two plots contained 13 new recruits, 57% of the total number of *A. rubrum* recruits observed over the entire study duration. We therefore cannot discount the possibility that our results are driven by abnormally high recruitment in two plots independent of insect presence. Indeed, insects exerted only weak, if any, effects on recruitment for both *A. rubrum* and *Fraxinus* spp., as insects increased recruitment by ~1 individual over the course of three years for both species. Given our small sample size and *a priori* hypothesis that recruitment should decrease in the presence of insects, it is likely that these patterns result from either Type S (wrong sign—i.e., finding a positive effect that should be negative) or Type M (i.e., overestimating the magnitude of an effect) errors (*Lemoine et al., 2016*). We therefore conclude that insects have little influence on the recruitment of tree species studied in this forest.

As predicted, insects increased mortality of *L. tulipifera*, but had little effect on the survival of most temperate forest species examined here, like *A. rubrum*, *Fraxinus* spp., and *F. grandifolia* (*Meiners, Handel & Pickett, 2000*; *Siemann & Rogers, 2003*). Many other temperate tree species, like *Quercus* spp. or *Prunus serotina*, are also resistant to herbivory or pathogens unless simultaneously subject to high densities of conspecific neighbors and, as a result, intense intraspecific competition (*Packer & Clay, 2000*; *Bell, Freckleton & Lewis, 2006*; *Burt et al., 2014*). Seedling density is therefore likely an important predictor of herbivore-driven mortality (*Paine et al., 2012*). Indeed, insects increased the mortality of only one species, *L. tulipifera*, which had the highest seedling densities of any species examined here. *Liriodendron tulipifera* produces exceedingly high numbers of seeds, releasing up to an order of magnitude more seeds m$^{-2}$ than most other coexisting species, except *A. rubrum* (*Greene & Johnson, 1994*; *Hille Ris Lambers, Clark & Lavine, 2005*). *Acer rubrum*, however, has a substantially smaller seed bank than *L. tulipifera* (*Hille Ris Lambers, Clark & Lavine, 2005*). As a result, *L. tulipifera* can produce large numbers of seedlings in any given year and likely experiences stronger density-dependent mortality than other tree species. In our study, *L. tulipifera* averaged 11.35 individuals m$^{-2}$, substantially lower than the densities of seedlings observed in either *Burt et al. (2014)* or *Bell, Freckleton & Lewis (2006)* (10–80 or 100–1,000 individuals m$^{-2}$, respectively) but similar to densities of *P. serotina* recorded by *Packer & Clay (2000)* (1–15 individuals m$^{-2}$). Our results therefore suggest that *L. tulipifera* is subject to density-dependent mortality even at relatively low seedling densities.

Likewise, insect damage did not affect growth of most species examined here, but rather reduced the growth of only *A. rubrum* and *L. styracifua*. Importantly, reduced growth was only observed in the youngest life stages; the impact of insects on growth of *A. rubrum* and *L. styraciflua* declined as gaps and seedlings aged. The declining effects of insects over time might be attributed to changes in insect abundance over time or, more likely, increased tolerance to insect damage by older seedlings. Indeed, seedlings alter their resource allocation patterns to maximize survival at any given demographic stage (*Zhang & Jiang, 2002*). Older plants have the carbohydrate reserves necessary to withstand loss of photosynthetic tissue (*Hanley, Fenner & Edwards, 1995*), thus making them more tolerant to herbivory. Additionally, foliar damage at our study site was generally low, ranging from

 

4 to 7% leaf area removal. This is, however, consistent with estimates of foliar damage on the same species (5–12%, *Siemann & Rogers, 2003*), across eastern North America (0–12%, *Adams & Zhang, 2009*), and within tropical forests (0–10%, *DeWalt, Denslow & Ickes, 2004*). Such increased tolerance to foliar damage likely mediates the declining impact of insects as gaps and seedlings age and might be responsible for the overriding effects of gap age on seedling growth.

As expected, plant life-history strategy appears to be an important determinant of the effects of insects. Early pioneer species, like *A. rubrum*, *L. tulipifera*, and *L. styraciflua*, all suffered reduced growth rates when exposed to insects. Herbivory may therefore help maintain forest diversity by decreasing the survival and growth of abundant, fast-growing seedlings: although *L. tulipifera* and *A. rubrum* comprised 36.3% and 8.5% of all tagged seedlings, the adult community is more diverse; *A. rubrum* and *L. tulipifera* comprise only 7.71% and 0.34% of adult trees in this study system (*Parker, O'Neill & Higman, 1989*). In contrast, *F. grandifolia* is a slow-growing, shade-tolerant, late-successional species that accounts for 20.77% of adult trees in our forest (*Parker, O'Neill & Higman, 1989*). In our study, *F. grandifolia* experienced the highest level of herbivory but remained largely unaffected by the pesticide, suggesting that species that resist or tolerate herbivory during the seedling stage can become dominant members of the overstory community. Variable susceptibility to herbivory among seedlings of temperate tree species therefore indicates that insects might influence trajectories of temperate forest succession by selectively preying on seedlings of specific species; abundant but palatable seedlings comprise less of the forest overstory than do abundant, non-palatable seedling species.

Here, we examined the role of insects at non-outbreak levels on temperate forest regeneration in treefall gaps. Although several studies have characterized the effects of insect herbivores on seedling growth and mortality in treefall gaps (*Norghauer & Newbery, 2014*), few track the effects of insects over multiple years. Interannual studies are especially important given yearly fluctuations in insect abundances, temperature, and rainfall (*Burt et al., 2014*; *Norghauer & Newbery, 2014*). We show that gap age is an important determinant of the effects of insects on tree seedlings; insects had weaker impacts in older gaps and on older seedlings. Importantly, our study examines the youngest life stage of tree seedlings (<15 cm tall), whereas many other studies of insect herbivory on tree seedlings use older, well-established seedlings that can withstand defoliation and loss of photosynthetic capacity (e.g., *Myster & McCarthy, 1989*). By conducting a multi-year study that examines the smallest seedling life stages, we demonstrated that insects have relatively weak effect on seedling survival and growth. Any effects of insect herbivore on forest succession are likely due to increased mortality of a single, dominant species: *L. tulipifera*. Long-term studies should examine whether or not insect herbivores can increase forest diversity by reducing the abundance of such competitively dominant tree species in temperate forests.

## ACKNOWLEDGEMENTS

We would like to thank N Buell, J Shue, A Shantz, W Drews, B Verrico, and L Maynard for their help in the field. The manuscript was greatly improved by the comments of two anonymous reviewers.

### Funding

This work was funded by a FIU FRSP grant to DEB and a Presidential Fellowship (FIU), Dissertation Evidence Acquisition Grant (FIU), a Dissertation Year Fellowship (FIU), and a USDA NIFA-AFRI Postdoctoral Fellowship (2016-67012-25169) to NPL. The funders had no role in study design, data collection and analysis, decision to publish, or preparation of the manuscript.

### Grant Disclosures

The following grant information was disclosed by the authors:
FIU FRSP.
Presidential Fellowship (FIU).
Dissertation Evidence Acquisition Grant (FIU).
USDA NIFA-AFRI Postdoctoral Fellowship: 2016-67012-25169.
Dissertation Year Fellowship.

### Competing Interests

The authors declare there are no competing interests.

### Author Contributions

- Nathan P. Lemoine conceived and designed the experiments, performed the experiments, analyzed the data, contributed reagents/materials/analysis tools, wrote the paper, prepared figures and/or tables, reviewed drafts of the paper.
- Deron E. Burkepile wrote the paper, reviewed drafts of the paper.
- John D. Parker conceived and designed the experiments, contributed reagents/materials/analysis tools, wrote the paper, reviewed drafts of the paper.

### Data Availability

The raw data has been supplied as a Supplementary File.

### Supplemental Information

Supplemental information for this article can be found online at http://dx.doi.org/10.7717/peerj.3102#supplemental-information.

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
