# Peer review of "Insect herbivores increase mortality and reduce tree seedling growth of some species in temperate forest canopy gaps"

_PeerJ, doi:10.7717/peerj.3102_

## Round 0.1 · original submission · Major Revisions

I agree with the reviewers that your manuscript addresses an important and interesting topic in ecology--how insect herbivory on seedlings influences plant community succession and composition. Your experimental approach contrasts with the much more abundant observational studies, thus representing an important advance in making strong inferences about cause and effect. And your application of this approach to temperate forests makes a relatively novel contribution to the plant ecology literature.

However, I do have a concern about the interpretation of the experimental manipulation. Throughout the manuscript text and figures, you describe the pesticide treatment vs control as no herbivory vs herbivory. But I don't think the pesticide would specifically kill the herbivorous arthropods. If so, please cite evidence for this. If indeed the pesticide treatment would affect all or most arthropods (presumably not internal feeders), wouldn't this experimental treatment have much more complicated effects than merely no herbivory vs herbivory? I suggest that you change the description of the experimental treatment to "pesticide vs control" throughout the results text, figures, and tables. In the discussion, you should interpret the effect of the pesticide treatment. If you are making the case that the predominant effect of the pesticide treatment is killing insect herbivores, then cite more evidence for this argument than you currently do. I do wonder if some of your surprising results may have resulted from unintended effects of the pesticide on arthropod predators.

In most other respects, I agree with the comments of the reviewers, and I expect you to address them thoroughly in a revision. If you choose not to follow the suggestion of a reviewer's comment, please justify your decision.

Reviewer 1 ·

Basic reporting

This is an interesting experiment and one that addresses an important question in plant community ecology and plant-insect interactions. I appreciated the experimental design approach of finding treefall gaps in forests and setting up long-term monitoring plots. I respect that the authors have used a hierarchical modeling approach to evaluate the results from an experiment that is highly nested. The inclusion of six tree species, a large range of data collected on the seedlings, an experimental manipulation, and the exclusion of deer are clear strengths of the study. The conclusion that abiotic factors related to gap dynamics in temperate forests probably exceed the importance of insect herbivores is a sound one given the authors’ results and what other ecologists have observed.

My concerns are as follows in order to have the paper pass:

The introduction is well cited, but it does not carry the reader from the primary research question to specific hypotheses and predictions. The study focuses on three categories of consequences of plant-insect interactions for seedlings, including growth, survivorship, and recruitment. It was unclear to me how these categories were chosen, how they have been used in past studies, and how changes to growth, survivorship, and recruitment would alter forest communities. Obviously these ecological impacts on young plants are important, but the specifics are necessary to make the connection between the plant-herbivore interaction and the resulting plant community-level patterns. The introduction would be greatly improved if it was rewritten to setup up three separate hypothesis such as the “seedling performance hypothesis,” “seedling mortality hypothesis,” and “sapling establishment hypothesis,” or some similar phrasing to this. Hypotheses set up in this way should clearly indicate the consequences for forest communities.

Experimental design

The experiment is well designed from the perspective of assessing changes to seedlings growth, mortality and recruitment.

However, there is no information on the insect herbivores responsible for herbivory or seedling predation. I do not question that the pesticide application works, and appreciate that there could be many multiple insect species driving the patterns observed in this system. However, in order to evaluate the validity of the results, the reader needs to have some information on what the typical insect community driving reduce seedling growth, increased mortality and recruitment. The experimental background needs to answer questions such as: Did the pesticide treatment actually reduce the abundance of herbivores? What is the duration of the pesticide application in each plot? What prevents insect herbivores from migrating into experimental plots once the pesticide application has worn off? What species of insects are most likely to be found in these plots feeding on seedling leaf tissue? Citing studies with the same plant species from this system would be sufficient, but some simple descriptive field data would be best.

Validity of the findings

While containing some interesting implications from the results, the discussion shares some of the same problems with the introduction. The discussion should be structured around the three primary hypotheses and end with a general comparison to other study systems. This would require some major changes similar to those proposed for the introduction.

The discussion ends abruptly. There should be a paragraph describing how the primary contribution of this work in addressing the knowledge gap the study intended to answer.

Some of the conjecture in the discussion is thought-provoking, but a few times the points lack sufficient background. See general line comments for further feedback on these points.

Additional comments

Line comments
Abstract –
The first few sentences of the abstract start too specific. The abstract should introduce natural enemies’ role in seedling mortality or tree recruitment as a central question in population and plant community ecology. Later the abstract should introduce that data was collected on recruitment, survivorship, and growth and introduce, at least a little bit, how these are indicators of plant population trends.

Line 30 – This sentence is a little awkward because it sounds like growth and abundance are properties of species. Here the authors should split up sentence to say that insect herbivores reduce the growth of individual plants and that herbivores can reduce the size of a plant population in a community.

Line 81: Can the authors cite a paper on the biome or ecoregion which establishes that the plant community used in this study is indeed typical to mid-Atlantic forests? It would also help to use a more specific but brief description of the ecosystem (upland, managed forest).

Line 84: Is this a plot size used in other studies? Please cite.

Line 88: Are these wire barriers around the entire experimental plots?

Line 149: The analysis is assessing the treatment effect, which was pesticide application. This is fine, but without a direct measure of the herbivore community, it would be incorrect to ascribe all the treatment effects to herbivore community alone. Be consistent with line 157.

Line 158-159: Last sentence can be removed and a simple statement about species level coefficients can be added to the previous sentence

Line 236-241: This is great information and should be front and center in the introduction.

250-251: What mechanism might lead to increased seedling recruitment in the presence of herbivores? Is there any suspicion this mechanism might be operating in this study system?

277-280: This is a bit of a reach. I suggest just attributing these patterns to increased tolerance to herbivory since the plant chemistry mechanisms are not included in the study nor cited earlier in the paper.

287: Remove italics on citation

295: Check “meters squared” formatting

298-300: This is an interesting conjecture, but one that has to be expanded a bit more. Why is density-dependence not operating in this study system?

309: It is unclear here if the tolerance of F. grandifolia to herbivores explains patterns of herbivory, or if it is merely that F. grandifolia is not palatable to herbivores. Is there a way to address this question in the analysis?

Fig 1-3 comments:

Figure title should be a concise summary of the figure and probably doesn’t need site information. The figure caption should also indicate which means and SE are significant/biologically important.

The authors can make the figure clearer by having pesticide treatment called “- herbivores” to indicate a manipulation and control “+ herbivores” to indicate herbivores being present.

Table 2: Format so numbers in cells line up like in table 3.

Reviewer 2 ·

Basic reporting

The authors investigate the impact of insect herbivores on deciduous tree seedlings in forest gaps using an insecticide treatment. Over three years, insecticide was applied to half of each gap, and tree recruitment, mortality, and growth were recorded. The authors conclude that insects have ‘an important but relatively cryptic’ role by reducing survival of one pioneer species and growth of others.

The manuscript is well-written in clear English. The introduction lays out a fairly good justification for the study, but the discussion isn’t linked to it as well as it could be. The opening paragraph of the discussion includes a comparison of tropical and temperate forest, which is the first time in the manuscript this has been brought up. Related to this, I don’t agree with the statement that “we know relatively little about” the role of insect herbivores in temperate forests.

Experimental design

The general experimental design is valid and takes advantage of new forest gaps. Pairing insecticide-treatment and control plots in each gap is an appropriate design, but the inability to include this pairing in analyses is a weakness of the study. The authors acknowledge this, but excluding the paired aspect of the study should be accompanied by some further justification that it would not affect the validity of the results. For example, if mortality/recruitment/growth in plots is highly correlated within each gap, this could indicate that variation among gaps is high, so accounting for this in models is necessary.

Similarly, measurements were taken in the same plots over multiple years, but it is not explained how or if the models account for this non-independence. The linear models presented in the appendix seem to treat each plot-year combination as independent, but my knowledge of HLMs is limited, so this non-independence may already be incorporated into the model evaluations. More details on this, at least in the appendix, would be helpful for readers (like me) without much HLM experience.

The analysis of recruitment in the manuscript only seems to include pesticide as a predictor variable (L192-196) but the appendix includes age and pesticide x age interactions. These are not reported in Table 2. This might just be a mistake in the appendix, but there should be clarity on the exact model structure used. If age and interactions were not used in these models, this should be justified because it seems unusual to use a different analyses for recruitment and mortality.

L228-230, I believe the growth effects in L. styraciflua are slightly misinterpreted. Pesticide is having a significant effect (I know, I’m using frequentist terminology) in year 1, indicated by the high Pr for pesticide. Year 3 and its interaction is not significant because it follows the same pattern as year 1 (evident in Fig. 3), with herbivores reducing growth. The high Pr for year 2 indicates that the overall pattern for year 2 is different (higher growth overall), and the high Pr for year 2 x pesticide interaction indicates that the effect of pesticide is different in this year (but not different in year 3). Thus years 1 and 3 follow the same reduced-growth pattern, with a pesticide effect only absent in year 3.

Validity of the findings

L254, Are there any hypothetical mechanisms by which insect herbivory could increase recruitment of A. rubrum? If the pattern is just driven by abnormally high recruitment in two plots, this same high recruitment would have occurred in the adjacent control plot in the same forest gap. This is a case where not being able to use a paired design weakens the inferential power of the experiment.

L284-287, these lines present the conclusion that seedlings are tolerant of herbivory, so herbivores have little effect. An important qualifier here is that the specific levels of herbivory experienced by these trees in this study system had little effect. 4-7% herbivory seems low to me, but it would be helpful to have comparisons to other temperate deciduous forests. If this particular site has relatively low abundances of herbivores, that would help make the results here more informative.

Any information on recent seed production would be helpful. Is it possible that strong effects are seen on L. tulipifera because it recently had a particularly high seed production year? This would agree with the conclusion that seedling density increases herbivore effects. If tulip trees generally produce more seeds than other species in this forest, that would support the generality of the results here.

Additional comments

-L43, can drop ref in parentheses
-L258, “these results and others” – what others are referred to here?
-appendix, 6th line of the first paragraph, inset “age” after “gap”
-appendix, add “3.” before “Recruitment Model”
-Table 1, maybe add in the number of gaps or plots each species was present in to illustrate how evenly or unevenly distributed each species was.

---

## Round 0.2 · accepted · Accept

I appreciate your careful and thoughtful responses to my comments and the reviewers' comments on the previous version of the manuscript. This version is much improved and ready for publication except for a few minor typological issues, which I have corrected in track-changes in the attached pdf file. I have notified the PeerJ staff to include these changes in the manuscript before publication.